# microRNAs in Subarachnoid Hemorrhage (Review of Literature)

**DOI:** 10.3390/jcm11154630

**Published:** 2022-08-08

**Authors:** Marianna Makowska, Beata Smolarz, Hanna Romanowicz

**Affiliations:** 1Charité—Universitätsmedizin Berlin, Corporate Member of Freie Universität Berlin and Humboldt-Universität zu Berlin, Department of Anesthesiology and Intensive Care Medicine, Augustenburger Platz 1, 13353 Berlin, Germany; 2Laboratory of Cancer Genetics, Department of Pathology, Polish Mother’s Memorial Hospital Research Institute, Rzgowska 281/289, 93-338 Lodz, Poland

**Keywords:** subarachnoid hemorrhage, miRNA, biomarkers

## Abstract

Recently, many studies have shown that microRNAs (miRNAs) in extracellular bioliquids are strongly associated with subarachnoid hemorrhage (SAH) and its complications. The article presents issues related to the occurrence of subarachnoid hemorrhage (epidemiology, symptoms, differential diagnosis, examination, and treatment of the patient) and a review of current research on the correlation between miRNAs and the complications of SAH. The potential use of miRNAs as biomarkers in the treatment of SAH is presented.

## 1. Subarachnoid Hemorrhage

### 1.1. Introduction

Subarachnoid hemorrhage (SAH) is a pathological, spontaneous extravasation of blood from a vessel into the subarachnoid space, located between the soft tissue and the arachnoid mater [1,2]. It is believed that in 80% of cases, it is caused by therupture of ananeurysm [3]. Other causes include inflammatory or non-inflammatory damage to the arteries of the brain, as well as sickle cell anemia and coagulopathies [3]. Poor treatment outcomes for patients with subarachnoid hemorrhage with aneurysmal subarachnoid hemorrhage (aSAH) are most often associated with delayed cerebral ischemia (DCI), which develops around 4–12 days after subarachnoid hemorrhage. DCI includes a number of pathological conditions, including clinical signs of cerebral vasoconstriction, ischemic neurological deficit, and asymptomatic, delayed cerebral infarction [3].

An aneurysm is an abnormal dilation of the artery caused by a disorder of the structure of the vessel wall and high arterial pressure. Aneurysms most often form in the places of bifurcation of large cerebral arteries, within the Circle of Willis. Blood flowing to the fork of the arteries under high pressure hits the weakened wall, causing its protrusion and the formation of an aneurysm. The larger the aneurysm, the greater the risk of subarachnoid hemorrhage [4].

According to studies, the occurrence of aneurysms among the population ranges from 1–6%. On average, it is assumed to be2%. However, in most cases, as many as 90% of cases, aneurysms are no larger than 1 cm and do not pose a high risk of bleeding [5]. They occur mainly in the anterior part of the arterial Circle of Willis, on the arteries: connecting the anteriorcarotid, internal carotid, middle brain, and anterior brain and less often in the vertebral-basal system [1]. Risk factors for the occurrence of an aneurysm include, among others, genetic tendencies—occurrence in the family, polycystic kidneys, Ehlers–Danlos syndrome. The other risk factors are being female, black, and/or over 50 years old, having hypertension, smoking, alcohol abuse, the use sympathomimetic drugs or other drug abuse (cocaine) [1,6,7].

### 1.2. Epidemiology

Data show that the incidence of subarachnoid hemorrhage is 6.67 per 100,000. These values range from 0.71 to 12.38 depending on thegeographical location [8]. Aneurysms are undisputedly the most common cause of SAH. The incidence of aSAH worldwide in 2007 was 9/100,000 people, with these values varying from country to country [9]. For example, in South and Central America, it was 4.2/100,000, while in Japan and Finland, it was 22.7/100,000 and 19.7/100,000 people, respectively [9]. The main problem, however, is not the incidence, but the level of mortality and persistent poor treatment outcomes. The WHO’s international team for monitoring trends in cardiological diseases (WHO MONICA stroke study), based on large-scale observations, presented results showinga 42% mortality rate within 30 days of the onset of the hemorrhage [10]. In addition, among surviving patients, 30% lose their independence and only 19–31% achieve the same quality of life as before the disease [9,10]. Another study described an annual incidence of aSAH ranging from 2 to 16 per 100,000 [11]. The incidence of subarachnoid hemorrhage increases with age. Most often, it affects people between 40 and 60 years of age. Many studies have reported a higher incidence of aSAH in women than in men [12,13,14,15]. A recent systematic review found the incidence of aSAH in women to be 1.4-times higher than in men [16].

### 1.3. Symptoms

The most common symptom of subarachnoid hemorrhage (80–97%) is a sudden headache of very high severity. It may be accompanied by vomiting and rapid loss of consciousness. Sometimes, loss of consciousness may be the first symptom of the disease, and after recovering consciousness, the patient always complains of a headache [17].

The second class of characteristic symptoms of subarachnoid hemorrhage ismeningeal symptoms, which are a consequence of the irritation of the meninges by the blood. However, it should be remembered that these symptomsdo not have to occur immediately after the onset of disease. They can occur up to 24 h after the onset of the disease [1,5].

A complication of subarachnoid hemorrhage may be hemiparesis, which can evolveinto paralysis, aphasia-type speech disorders, and symptoms of damage to the cranial nerves—the optic and olfactorynerves (Figure 1).

In addition, in acute, complicated subarachnoid hemorrhage, symptoms involvingother systems may occur. In 10% of cases, Terson syndrome appears, characterized by hemorrhage into the inside of the eyeball as a result of an increase in intracranial pressure [5,18].

Disorders of the cardiovascular system are often observed in patients with subarachnoid hemorrhage from a ruptured aneurysm. Shortly after the onset of extravasation, there is a sharp increase in natriuretic peptide type B (BNP) and cardiac troponin type I and T (cTnI, cTnT). The data indicate a close relationship between troponin and BNP concentrations and the occurrence of myocardial pathologies as a result of aSAH. The genesis of heart disorders is associated with the ejection of endogenous catecholamines [19]. A link has been shown between the massive release of catechol amines and circulatory disorders, in the early stages after aSAH [15].

Further symptoms are disorders of waterelectrolytes—cerebral salt loss syndrome. Thesearecharacterized by hyponatremia and dehydration, resulting from the overproduction of natriuretic hormones. The second symptom is Schwartz–Bartter syndrome—characterized by hyponatremia along with the hypoosomolarity of urine [5,20].

In conclusion, the main symptoms of subarachnoid hemorrhage are:-Sudden, severe headache;-Meningeal symptoms;-Nausea and vomiting—in a retrospective review of 109 patients, Fontanarosa et al. reported that 77% of patients with aSAH had nausea or vomiting [21];-Photophobia;-Disturbances of consciousness up to and including loss of consciousness;-Neurological deficits;-Seizures of an epileptic nature [5].

In 1968, Hunt and Hess proposed a scale that is still valid today, which allows assessing the degree of severity of SAH (Table 1) [22].

An alternative scale was presented by the World Federation of Neurosurgical Societies (WFNS) (Table 2). It is largely based on the Glasgow coma scale and the presence of movement disorders.

### 1.4. Differential Diagnosis

Subarachnoid hemorrhage should be differentiated fromthe following diseases:-Venous cerebral thrombosis—a disease whose predominant symptoms are headache, impaired consciousness, and swelling of the optic nerve disc; focal symptoms with various motor and sensory deficits and defects, as well as speech and vision disorders may also occur [23];-Meningitis—the main symptoms are slowly developing headache, fever, neck stiffness, and impaired consciousness; vomiting, convulsions, photophobia, hypersensitivity to sounds may also occur [24];-Migraine—manifested by a severe half headache; it can occur with an aura or without. The peak of the disease is observed in the age group of 30–50 years [25].

### 1.5. Interview

Aninterview is carried out if the patient is conscious, and the patient mayreport a sudden, very strong, splitting, headache. Thismay be accompanied by photophobia, vomiting and nausea, as well as anxiety and agitation. The first symptom of the disease, especially with massive bleeding, may be a sudden loss of consciousness or an epileptic seizure [26]. A thorough interview is important tomake the right diagnosis and for further management of the patient. It is important to know whether the patient has suffered from a headache of a similar, but less severe nature in the last few weeks. Thiscould be indicative of a smaller hemorrhage announcing the currentevent. The circumstances of the onset of pain are quite important. They can be associated with stress, defecation, sexual intercourse, and physical exertion, but are mostly associated with daily activities [5].

It is also important to collect an anamnesis of past and chronic diseases, including hypertension. It is necessary to find out whether the patient is being treated for them, what drugs are being used, whether he/she takes them regularly, and what the effect is. Important clues may be information related to cigarette smoking and alcohol abuse. It is also worth asking about other drugs and psychoactive substances, with an indication of those that stimulate the sympathomimetic system, e.g., cocaine, because thisrepresentsa significant risk factor for subarachnoid hemorrhage [6,27]. Oneshould also ask about the occurrence of subarachnoid hemorrhage in the family. Alternatively, information about sudden deaths should be obtained.

### 1.6. Diagnosis and Treatment

The priority is to transport the patient to the appropriate hospital unit as quickly as possible.

Patient management will be symptomatic. During transport, it will be necessary to secure peripheral intravenous punctures. In case of vomiting, metoclopramide should be administered. If symptoms of intussusception appear, dehydration drugs (Mannitol 20%, 250 mL) should be administered, and the patient may also be ventilated. Transport of the patient should take place in a supine position, so that recurrenceof hemorrhage does not occur [28]. Hospital management consists ofconfirming the diagnosis of subarachnoid hemorrhage with the help of imaging diagnostics. First, a tomography of the head without a contrast agent is performed. This is the gold standard in diagnosing subarachnoid hemorrhage. The sensitivity of this test in the first 3 days after the onset of symptoms is almost 100%. Then, it drops to reach 50% in the range of 5–7 days after the onset of symptoms. During the test, hyperdense fluid reservoirs aresought [28,29]. If noblood isvisible, but the suspicion of subarachnoid hemorrhage is high, then a lumbar puncture is performed to examine the cerebrospinal fluid. The earliest this examination can be performed is6 h after the onset of symptoms. The examination looks for xanthochromia—a yellowish coloration of the liquid after centrifugation [27]. Magnetic resonance imaging (MR) is also performed in the sequence of FLAIR, T2, and density of PD-dependent protons, which allows visualizing subarachnoid hemorrhage, sometimes identifying aneurysm and detecting other vascular pathologies [1,5]. In order to find the cause of bleeding and determine its location, vascular examinations of angio-CT and angio-MR areperformed. To assess the severity of bleeding in computed tomography (CT) imaging, the Fisher Scale (Table 3) is used. Thisalso allows estimating the risk of vascular spasm.

However, subtractive angiography of cerebral vessels has the greatest sensitivity and specificity in the diagnosis of ruptured aneurysms [1,5,27,29]. The main goal of treating subarachnoid hemorrhage is to avoid repeated bleeding. In the case of surgery foraneurysms, thisconsists ofsurgically excluding the aneurysm from the circulation. Inoperable aneurysms are embolized by an interventional radiologist. The surgery involves putting a clip on the neck of the aneurysm in order to mechanically cut off the blood supply to the bag and, at the same time, maintain proper blood flow through the vessel. The second method is the introduction of a coil into the lumen of the aneurysm. Thisis a form of spring that allows retaining the blood in the aneurysm, as well as facilitates the coagulation process on its surface. The procedure is performed using a catheter inserted through the femoral artery and until arriving at the damaged vessel [5]. The third method is the so-called wrapping. It consists ofcovering the aneurysm with muscles or a plastic mass. The fourth is the formation of a form of bypass over the place where the aneurysm occurred. These types of vascular bridges are performed usingvascular grafts from the saphenous vein or temporal artery. In patients in whom interventional treatment is impossible, conservative treatment is carried out. Thisconsists oflimiting motor activity and using sedatives and/or analgesics, combating cerebral edema and preventing vascular spasms [27].

One of the complications of subarachnoid hemorrhage is ischemic stroke. It appears between 4–10 days after the onset of the disease and is caused by vascular spasms. Vascular spasms appear as a result of the contact of extravasated blood with the walls of the vessels of the brain. Theselead to vasoconstriction and, secondarily, can lead to ischemic stroke [30]:-To avoid this type of complication, the 3H rule is used:-Hypervolemia—patients with subarachnoid hemorrhage tend to fall into hypovolemia, associated with improper cerebral flow;-Maintaining increased blood pressure (hypertension);-Hemodilution [30].

However, caution must be exercised. Increased fluid supply and increased blood pressure may cause the recurrence of bleeding [31].

Another factor that counteracts vasoconstriction is the supply of a calcium channel blocker—nimodipine. It is administered at a dose of 60 mg every 4 h for 21 days after the incident has occurred. The use of nimodipine reduces the risk associated with the occurrence of ischemia in the brain [32].

The administration of magnesium, which is a physiological calcium antagonist, works in a similar way. Like nimodipin, it prevents neurological deficits [32]. Currently, to counteract the occurrence of vasoconstriction, it is suggested to administera colloidal solution. In addition, the introduction of a catheter into the subclavian vein should be considered in order to maintain venous pressure within 8 and 12 mmHg and arterial pressure within 20 and 40 mmHg ofthe mean value calculated from the formula. Hypoxemic therapy, fluid supply—no less than 3 L per 24 h—and treatment of hyponatremiamay also be used [31].

In a fifth of patients, hydrocephalus may appear. It is associated with a hindered outflow of cerebrospinal fluid from the ventricles. This is a factor conducive to a worse prognosis. This is due to the fact that there is an increase in intracranial pressure, which, in turn, can lead to intussusception of the brain. To reduce cerebral edema, osmotically active agents are administered and ventilation is used. A procedure to reduce hydrocephalus is ventricular drainage [33].

Another complication of subarachnoid hemorrhage is epileptic seizures. Like the occurrence of hydrocephalus, this factor also worsens the prognosis. As for the treatment of seizures alone, there is no consensus. Rather, symptomatic and ad hoc treatment is indicated [34]. The prognosis after the onset of subarachnoid hemorrhage is rather poor. Mortality ranges from 32% to 67% [35]. Forty-five percent of patients survive with severe neurological deficits, up to 20% with various types of disability [35]. Timely diagnosis, appropriate intervention, proper diagnosis, and treatment are crucial for the survival and subsequent quality of life of the patient. Previous studies have shown that environmental exposures and genetic predisposition play a role in the susceptibility to SAH, with the estimated heredity being around 40% [36].

With significant advances in therapy, SAH remains a very difficult condition associated with a high socioeconomic burden [37,38]. SAH is a disease that needs to be treated immediately. In the treatment of SAH, an in-depth understanding of the molecular mechanisms is necessary. In addition, early screening and early active treatment and prevention of SAH help reduce the mortality and disability rates for patients. With the development of bioinformatics, gene expression profiling has been widely used to identify biomarkers for the diagnosis and treatment of SAH [39,40]. Recently, attention has begun to be paid to non-coding RNAs—microRNAs (miRNAs)—although many other classes of experimentally identified ncRNAs with various lengths and characteristics have been reported in the literature (endo-siRNAs, piRNAs, snoRNAs, sdRNAs, tiRNAs, moRNAs, circRNAs, lncRNAs, lincRNAs, T-UCRs). miRNAs are known to regulate many processes such as transcription, translation, regulation of cell differentiation, and cell cycle [41]. Studies indicate that non-coding RNAs play an important role in intracranial aneurysm (IA) and, thus, aneurysmal subarachnoid hemorrhage [42].

## 2. microRNAs

The first gene encoding miRNA (lin-4 miRNA) was described as early as 1993 [43,44]. The groups of Ambros and Ruvkuna, studying the genes that control the passage of Caenorhabditis elegans through successive larval stages, determined that the expression of lin-14, one of the most important regulators of the early stages of nematode development, is inhibited by short transcripts of lin-4 (22 and 61 nt). It was found that these molecules do not encode proteins, but contain sequences complementary to fragments of 3′UTR (untranslated region) mRNA lin-14 and may participate in the regulation of the expression of this gene through a mechanism unknown at the time.

However, since lin-4 has no homologues in other organisms, the biological significance of the discovery made in the laboratories of Ambros and Ruvkuna remained unnoticed for a long time. Although gene silencing in plants was observed as early as 1990 [45] and, in the following years, similar effects were also described in fungi of the genus Neurospora [46] and in Drosophila melanogaster [47], these phenomena were not associated with small RNA molecules.

The breakthrough came in 1998, when Fire and Mello’s team published results explaining the basis of a mechanism the researchers called RNA interference. It was then proven that simultaneous administration of sensible and arbitrary RNA to C. elegans cells leads to 10-times more efficient gene silencing than when using a single strand, and only a few RNA molecules per cell are enough to produce this effect. The significance of this discovery was recognized by the awarding of the Nobel Prize in Physiology or Medicine in 2006. In the following years, it was established that the RNA initiating the interference process is cut into short fragments in the cell with a length of 21–23 nt. [48]. At the same time, another short 21 nt let-7, which inhibited the expression of rope genes, was identified in C. elegans cells [49]. The demonstration that let-7 homologues are common in living organisms [50] and are part of a separate mechanism for regulating gene expression, which isRNA interference (RNAi) [51], initiated a real miRNA revolution. Since then, great progress has been made in understanding the role that these molecules play in living organisms: thousands of genes encoding them have been identified, and biogenesis pathways and mechanisms of miRNA function have been described [52].

### miRNA Biogenesis

microRNAs (miRNAs) are endogenous non-coding RNAs composed of 18–22 nucleotides that regulate gene expression at the post-transcriptional level by interacting with the non-translatable 3′ (3′UTR) regions of the target mRNAs. Genes for miRNAs are a part of the protein-coding sequences (introns or exons) or function as stand-alone transcription units [53,54,55]. They are transcribed with the participation of RNA polymerase II or III to pri-miRNA, with a length of up to several thousand nucleotides [56,57]. They have intramolecular regions of complementarity, as a result of which, they form interconnected structures of hairpins. At the end of 5′ pri-miRNA, 7-methylguanosine is attached; the 3′ end is polyadenylated [58].

Two enzymes from the ribonuclease type III family are involved in the maturation of pri-miRNA: Drosha and Dicer. Drosha and the protein Di George syndrome critical region gene 8(DGCR8) form a functional enzymatic nuclear complex called a microprocessor [59]. It is postulated that the DGCR8 protein participates in the diagnosis of pri-miRNA even before it binds to Drosha. It binds the single-stranded ends of pri-miRNA and directs the catalytic domain of Drosha ribonuclease to the hydrolysis of transcripts to fragments with a length of ~60–100 nucleotides, the so-called pre-miRNAs [60]. Pre-miRNAs also adopt the structure of hairpins, and in this form, with the participation of the exportin transporter 5 interacting with the Ran protein, they are transferred to the cytoplasm [61].

In the cytoplasm, pre-miRNA undergoes further maturation with the participation of the Dicer enzyme associated with the Argonaut (Ago) protein andtransactivating response RNA-binding protein (TRBP) [62]. Dicer recognizes both double-stranded and single-stranded regions of unpaired ends of pre-miRNAs and then hydrolyzes the molecule at the base of its loop [52]. The result is double-stranded RNA with a length of ~20 nucleotides (miRNA–miRNA duplex*). One of the duplex strands, the passenger, degrades; the other, the leading one, is a functional molecule. In some cases, both strands are incorporated into the microRNA-induced silencing complex (RISC) [63,64].

There are also known miRNAs, called non-canonical, thatare formed by omitting some elements of the miRNA biogenesis pathway described above, usually Drosha ribonuclease and the DGCR8 protein and, less often, Dicer ribonuclease [65]. For example, in Drosophila melanogaster, Caenorhabditis elegance, and later, also in vertebrates, it was observed that the source of miRNA may be introns [66,67]. They are formed with the participation of spliceosomes, which in this case replace the activity of Drosha and DGCR8. The source of miRNAs can also be small nucleolar RNA (snoRNA). snoRNAs participate in RNA processing, consisting mainly of rRNA modifications and, less often, snRNA and mRNA and are involved in the maturation of pre-rRNAs and pre-mRNAs. Relatively recently, it has been observed that they can also form complexes with Ago proteins. An example is the human snoRNA, ACA45, which resembles two pre-miRNA clips fused together. They are recognized by Dicer, and its product participates in silencing gene expression by destabilizing the mRNA and regulating the initiation of its translation [68,69]. It was later discovered that other endogenous RNAs resembling pre-miRNA clips can also be treated with Dicer and be a source of functional miRNAs, for example miR-320 and miR-484 [70]. miRNAs may also be a by-product of the maturation or degradation of tRNA [71]. They are formed with the participation of tRNase Z or Dicer, independently of Drosha and DGCR8. For the first time, they were observed in cells infected with HIV. Another example of a non-canonical miRNA biogenesis pathway is the maturation of miR-451, which is formed independently of Dicer ribonuclease [72].

Pri-miR-451 is processed with the participation of Drosha/DGCR8 into a pre-miRNA with a length of only 18 nt, which is not a substrate for Dicer ribonuclease, but is directly incorporated into the RISC complex [73]. The source of miRNAs can also be viral transcripts. To date, more than 300 miRNAs of viral origin are known [74].

It happens that they mature independently of Drosha, but always with the participation of Dicer ribonuclease [75,76]. Significant sequence similarities have been noted, particularly in the “seed” region, viral miRNAs, and host miRNAs, for example SFVagm-miR-S4 and has-miR-155; and SFVagm-miR-S6 and hsa-miR-132 [77,78]. Viral miRNAs inhibit the immune response of the cell, thus enabling their longer survival in the host organism [79].

The scheme of miRNA biosynthesis is shown in Figure 2.

Mature miRNAs are incorporated into the RISC complex, the main element of which is proteins of two families, Argonaute and GW182 [80]. miRISC binds target transcripts [81], for which, according to the canonical model of action, full complementarity of the “seed” region of the miRNA (2–7 nucleotide) and the target mRNA within its 3′UTR is required. Later studies showed that non-canonical and even non-evaporation interactions are acceptable between the “seed” region of miRNA and mRNA [82].

Moreover, it has been proven that non-seed nucleotides, in particular 9–12 miRNA nucleotides, may also be involved in mRNA binding [83]. In addition, miRNA binding sites are not, as originally thought, only in 3′UTR mRNA, but also in 5′UTR [84] and even its coding regions [85]. Binding miRISC to the target mRNA inhibits translation [86,87,88,89,90,91,92]. The function of miRNA is not limited to inhibiting protein biosynthesis. They can induce the translation of selected mRNAs [93].

miRNAs affect the course of processes of fundamental importance for the proper functioning of the body. These processes include cell division, proliferation, differentiation, cell apoptosis, as well as the formation of blood vesselsand, finally, cancer.

Research on the properties of miRNAs and their possible applications is still ongoing and has great prospects. One of the aspects of their use is the issue related to the occurrence of subarachnoid hemorrhage, which 80% of the time is associated with IA. An intracranial aneurysm is an abnormal focal dilation of an artery in the brain that results from a weakening of the inner muscle layer of the blood vessel wall. IAs represents the most common cause of non-traumatic subarachnoid hemorrhage. Despite technological advances in the treatment and use of new IA diagnostic methods, they still pose a significant risk of mortality and disability. In addition, due to a poor understanding of the mechanisms of SAH, the current diagnosis and treatment of SAH may be inconsistent and/or ineffective [94,95].

Numerous miRNAs (called circulating miRNAs) have been detected in biological fluids of the human body such as blood and cerebrospinal fluid (CSF). The expression profile of circulating miRNAs corresponds to the percentage of cells in which they are modified. miRNAs are secreted according to the physiological or pathological states of these cells. Circulating miRNAs can be secreted from cells into human biological fluids in extracellular vesicles. miRNAs can bind to the Ago2 protein. As a result of this combination, they are resistant to RNAses. For this reason, circulating miRNAs could become potential biomarkers for IA and SAH [96,97].

## 3. miRNA in Neurological Disorders

About 70% of the human miRNAs known so far occur in the brain. They are thought to regulate the expression of about half of human genes, and virtually every cellular pathway is influenced by them. They are therefore an excellent target for the search for new therapeutic methods for diseases of the central nervous system.

Several miRNAs (miR-9, miR-124a/b, miR-135, miR-153, miR-183, and miR-219) have been shown to be specifically expressed in differentiating neurons, suggesting that these miRNAs act as effectors in neuronal processes. miRNAs aretissue-specific and may play an important role in the brain at the subcellular level [98].

So far, it has been proven that an increasing number of miRNAs is crucial for the pathogenesis of neurological diseases. The discovery of miRNAs has expanded the potential of diagnostic markers and therapeutic targets for human diseases, including neurological diseases such as Alzheimer’s disease (AD), Parkinson’s disease (PD), and Huntington’s disease (HD). Emerging evidence suggests that the most common sporadic forms of AD and PD may be due to the increased expression of genes encoding the precursor protein beta-amyloid (APP), tau (AD), and alpha-synuclein (PD). Therapeutic methods are sought to reduce the expression of alpha-synuclein, APP, and tau in neurons [99].

Recent studies using cell cultures and animal models have shown the ability to express miRNAs in neurons using lentiviral vectors configured to produce short RNAs with a hairpin structure. For example, the torsin A protein associated with dystonia disease DYT1 was suppressed in primary neuronal cultures using recombinant feline immunodeficiency virus or shRNA-expressing lentivirus vectors. Lentivirus-mediated RNA interference is also used to suppress alpha-synuclein levels in neurons in culture and in vivo in the rat brain. Lentivirus-based gene therapies for a range of diseases are at different stages of clinical development, and RNA-based neurodegenerative disorder therapies are expected to be finally tested. Both changes in miRNA expression and disruption ofmiRNA function by mutations in 3′UTR play a role in neurological dysfunction. Manipulation of endogenous miRNAs altered as a result of disease pathology or the introduction of artificial miRNAs through the administration of oligonucleotides or technology based on viral vectors can provide effective treatment of these conditions. In addition, evidence that miRNAs mediate the action of antidepressants may provide new and more effective treatments for depression and related diseases.

Intracranial aneurysms arethe most common type of cerebral vascular defect. IAs represent the most common etiology of nontraumatic subarachnoid hemorrhage.miRNAs are involved in the pathogenesis of IA. Depending on various human diseases, including cardiovascular diseases, the expression profile of circulating miRNAs changes [100]. Changes in the expression of circulating miRNAs in IA have beendetected, suggesting that they may constitute a group of new diagnostic and prognostic markers.The mir-143/145 complex is intensively analyzed in the pathophysiology of cardiovascular diseases.The relationship between plasma concentrations of miR-143/145 and serum metalloproteinase 9 (MMP-9) was investigated in cases with unruptured or ruptured IA [101].Plasma levels of miR-143/145 were significantly lower in patients with IA than in the healthy control group. Serum levels of MMP-9 were significantly higher in patients with ruptured IA compared to patients with unruptured IA and healthy subjects. There was no significant correlation between plasma miR-143/145 levels and serum MMP-9 levels. It has been suggested that lower plasma levels of miR-143/145 may be associated with the formation of IA. Higher serum levels of MMP-9 may be correlated with the rupture of IA. Serum miR-143/145 levels were shown to be significantly reduced in patients with IA compared to controls. The cluster miR-143/145 can therefore be involved in the formation and progression of IA.

miR-155 participates in the process of vascular remodeling. This is very important for complex adaptive responses in cardiovascular disease, including IA [102,103]. A potential target for mir-155 is matrix metalloproteinase 2 (MMP-2).The possible binding site is located in the 3′-UTR MMP-2 [102]. An increase in serum relative expression of miR-155 was demonstrated in patients in the non-IA group compared with the IA group with rupture. A change in serum miR-155 expression can therefore be used to predict IA rupture.

Plasma expression of miR-15a-5p, miR-34a-5p, miR-374a-5p, miR-146a-5p, miR-376c-3p, miR-18b-5p, miR-24-3p, and miR-27b-3p was significantly altered in patients with aSAH [104]. miRNA levels were significantly deregulated only in cases of aSAH and not in patients with SAH for other reasons. Circulating miR-146a-5p and miR-27b-3p were associated with clinical outcomes in patients with aSAH. These miRNAs may play a key role in IA pathology by regulating many of thesignaling pathways associated withinflammation. These miRNAs interact with 2896 genes and areinvolved in transformative growth factor-β, mitogen-activated protein kinase, focal adhesion, and the phosphoinositide kinase 3/protein kinase B signaling pathway.

## 4. Circulating miRNAs in SAH

Intracranial aneurysm (IA) causes many systemic effects and strongly affects the functioning of the immune system. The molecular mechanisms that guide the effects of subarachnoid hemorrhage remain to be clarified to improve the treatment of patients with IA. One of the systemic effects of IA is changes in the transcriptome of peripheral blood cells.

In the case of intracranial aneurysm, several immunological, genetic, and inflammatory biomarkers, as well as factors related to angiogenesis or cell adhesion have been studied [105]. miRNA disruptions are associated with a variety of neurological diseases, including cerebral hemorrhage.Recent studies indicate circulating miRNAs as non-invasive biomarkers for IA and aneurysmal SAH (Table 4). They can be more sensitive and accurate in the diagnosis and prognosis of aneurysm rupture, as well as in assessing the effects after its rupture IA.

Circulating miRNAs can becharacterized by a non-invasive detection technique. They are very stable and have a long half-life in the sample. Depending on the tissue and disease, the circulating miRNA profile may show high specificity. Changing the expression profile of circulating miRNAs canallow detecting a high risk of IA rupture. Circulating miRNAs provide a promising tool for faster and more accurate identification and subtypes of stroke (distinguishing aSAH from spontaneous SAH). Protein biomarkers are detected in bioliquids only when a significant proportion of the damage has already occurred. In the case of miRNAs, they can be detected in the early stages of development. The small size and chemical composition of circulating miRNAs simplifies the analysis. miRNAs are less complex molecules than most biological molecules in the blood or cerebrospinal fluid [100,116,117,118].

Studies have been undertaken that have analyzed circulating miRNAs for IA and aSAH in order to better understand the function of miRNAs in this pathology and potentially introduce circulating miRNAs as new biomarkers in clinical practice.

The research by Sheng et al. showed that high levels of circulating miR-1297 expression in the serum of patients with aSAH were associated with a poor prognosis [109]. Serum miR-1297 had great power to distinguish patients with aSAH from healthy controls, especially 72 h after aSAH. In addition, in patients with aSAH, serum miR-1297 levels negatively correlated with aSAH severity assessed by the WFNS grade [109]. In addition, the researchers showed that serum miR-502-5p levels in patients with aSAH increased after the first 24 h after aSAH compared with healthy control groups and peaked after 168h or 7 days. It is worth noting that from Day 7 to Day 14, the level of expression of circulating miR-502-5p decreased [106].

In another study, serum expression levels of circulating miR-502-5p, miR-4320, and miR-1297 were significantly increased in patients with aSAH compared to the control group [108]. They also found that the expression of miR-502-5p and miR-1297 levels in serum in aSAH patients with a higherWFNSgradewas higher than in those with a lower WFNSgrade. Eight circulating miRNAs have been identified that may serve as a biomarker candidate for IA [104]. The study found that these were 3 low-expression miRNAs (miR-15a-5p, miR-34a-5p, and miR-374a-5p) and 5 elevated expression miRNAs (miR-146a-5p, miR-376c-3p, miR-18b-5p, miR-24-3p, and miR-27b-3p). The expression levels of eightcirculating miRNAs were significantly deregulated only in the cases of aSAH and not in patients with SAH arising from other causes. Hereby, circulating miR-146a-5p and miR-27b-3p were associated with clinical outcomes in patients with aSAH [119].

Powers et al. showed that miR-92a and let-7b may have pleiotropic effects on brain damage and brain response to trauma [113].

The following causes of death in patients with IA are known. This is a direct effect of leakedblood, or SAH, hydrocephalus, and late cerebral ischemia (LCI), which results from vasoconstriction [120]. The first two causes are effectively treated by improving the microsurgical technique of aneurysm surgery and hydrocephalus control methods. The main complication remains vasoconstriction and the associated LCI [121]. Due to the fact that cerebral vasospasm is a temporary complication, relatively rapid detection and rapid intervention can prevent LCI. It has been shown that the mechanism of vasoconstriction after aSAH consists ofa significant reduction in the level of nitric oxide (NO). NO as a mediating factor of vasodilation is crucial for regulating proper vascular tension.NO is synthesized by the enzyme nitric oxide synthase (NOS). The causal role of NO is evidenced by the disappearance of NO synthase during vascular onset during contraction. In addition, there is the destruction of NO by hemoglobin, which is released from blood clots in the subarachnoid space [122]. A negative regulatory relationship between miR-24 and NOS3 has been demonstrated [123]. A negative correlation occurred between the mRNA expression levels of miR-24 and nitric oxide synthase 3 (NOS3) in vascular tissue samples of patients with aSAH. In the above study, transfection with an miR-24 inhibitor increased the level of NOS3 expression. When transfection with synthetic analogues of miR-24 (mimicking) or siRNA NOS3, the level of NOS3 expression decreased in vitro. miR-24 expression levels were shown to be elevated in patients with aSAH with vasoconstriction compared with patients without contraction. Reverse results were observed for NOS3. It is assumed that aSAH may cause changes in miR-24 expression in cerebral arteries, which, when released into the bloodstream, may act as new biomarkers to assess the risk of cerebral vasoconstriction and/or LCI. However, this work must continue. As a result of the NGS technique, profiling of circulating miRNAs in whole venous blood was performed between patients with vasoconstriction with aSAH and patients who did not develop vasoconstriction [124]. Elevated levels of miR-3177-3p were associated with the risk of vasoconstriction in patients with aSAH, while an increase in miR-3177-3p levels was accompanied by a decrease in the expression of the lactate dehydrogenase A (LDHA) gene.Studies have shown that LDHA expression in cerebrovascular endothelial cells is affected by hypoxia, a key regulatory mechanism involved in vasoconstriction [114].

The data provide evidence that circulating miR-3195, miR-4788, and miR-1914 may be sensitive, good biomarkers for dynamic monitoring of aSAH development [112,115,125].

Due to the use of the latest molecular biology techniques (NGS, qRT-PCR), the expression of eight miRNAswas detected in patients with aSAH. Three miRNAs (miR-146a-5p, miR-589-5p, and has-miR-941) showed increased expression. Five miRNAs were shown to be reduced in expression (let-7f-5p, miR-126-5p, miR-17-5p, miR-451a, and miR-486-5p) [126]. In addition, circulating miR-486-5p showed the highest levels of expression and did not correlate with poor neurological status. It is suggested that the above circulating miRNAs can only be represented in aSAH, taking part in the pathogenesis of IA. miR-146a-5p is known to be involved in the regulation of inflammation and the regulation of innate immune responses in monocytes and macrophages.In addition, there was a relative increase in circulating expression of miR-146a-5p, miR-21, and miR-221 in cerebrospinal fluid in patients with aSAH with LCI compared with patients without LCI. These studies may contribute to the identification of new biomarkers of clinical relevance [127].

A microarray study in patients after aSAH in the cerebrospinal fluid showed that 256 circulating miRNAs were expressed differently [128]. Among these miRNAs, eighteen (miR-301a-3p, miR-378d, miR-137, miR-320e, miR-346, miR-514-15:00, miR-521, miR-624-3p, miR-708-5p, miR-1244, miR-2117, miR-4521, miR-302a-3p, miR-548I, miR-566, miR-27a-3p, miR-516a-5p, and miR-1197) showed a significant change in expression.

Lower levels of circulating miR-451a expression were observed in patients who experienced aSAH with vasoconstriction, compared to patients who experienced aSAH without vasoconstriction [129].

Recent studies have shown that RNAs (such as JMJD1C-AS1, LINC01144, hsa-miR-510, TLR4, ADRB2, TGFBR3) may be potential biomarkers for predicting SAH [130]. Studies in rats have shown that LINC01144-hsamiR may play a role in SAH by regulating the expression of ADRB2 and TGFBR3 [131]. ADRB2 encodes the beta2 adrenoceptor, while TGFBR3 is the beta receptor for transforming growth factor (TGF). Inhibition of the TGF-β1/Smad/ CTGF pathway prevented the development of hydrocephalus after SAH [132]. Effects onTGF-β1 in human mesenchymal stem cells derived from umbilical cord may attenuate SAH-induced chronic hydrocephalus, inflammatory cytokine expression, and other behavioral changes [133].

miRNAs have a significant regulatory effect on the destruction of blood–brain barrier integrity and the neuroinflammatory response secondary to SAH. In studies in cerebral hemorrhage rats, serum miR-126-3p levels were lowered and miR-126-3p overexpression significantly reduced the permeability of the blood–brain barrier in the bleeding area (BBB) [134]. miR-27a-3p inhibits neuronal apoptosis and microglial activation in hemorrhagic lesions in SAH rats and relieves BBB damage and neurological disorders [135]. miR-31-5p has a pro-inflammatory effect and worsens endothelial cell damage [136]. circARF3 suppresses the expression of miR-31-5p as competitive endogenous RNA (ceRNA). In terms of action, circARF3 can significantly weaken the destruction of BMEC integrity by targeting miR-31-5p and promoting microglial inflammation, further deepening our understanding of the protective effects in SAH mediated by circARF3 [137].

It has been suggested that miRNA may help doctors predict the risk of SAH and intervene as soon as possible. miRNA studies should focus on the main and avoidable causes of death and complications in patients with aSAH: cerebral vasospasm and/or LCI.

We still do not know the actual relationship between miRNA and cerebral vasospasm and/or LCI. It is interesting to note that for some patients suffering from cerebral vasospasm, LCI may develop, while in other patients with vasoconstriction, LCI does not occur.Therefore, the process of identifying new miRNAs may allow understanding complications after aSAH, as well as developing new interventional and therapeutic biomarkers.

## 5. Exosomal miRNAs in aSAH

Exosomes are vesicles with a diameter of 40–100 nm, surrounded by a membrane consisting mainly of proteins and lipids. They are released by most cells into the extracellular matrix in the process of exocytosis, which occurs as a result of the fusion of multivesicular bodies (MVBs) with the cell membrane. They transport both proteins and nucleic acids, including fragments of DNA, mRNA, miRNA, and other non-coding RNA [138,139].

Scientific research shows that miRNAs contained in exosomes are more stable and less susceptible to degradation than those found in the cytoplasm of cells. Transported in the form of vesicles, they are protected from harmful factors of the extracellular environment thanks to the lipid barrier. Exosomes can deliver multiple miRNA molecules to target cells simultaneously, thereby regulating numerous signaling pathways in them, making them an extremely attractive way of communicating between donor and recipient cells [140].

Literature data indicate a direct role for exosomal miRNAs in the development and progression of IA [141,142,143]. Twenty-fivecirculating miRNAs derived from exosomes may be useful as biomarkers for differentiating stroke subtypes, including aSAH [144]. Continuing clinical trials of circulating miRNA panels may improve care for stroke patients. miRNA panels can be converted into a point-of-care (POC) test that can be used in the clinic to help diagnose and segregate patients.Studies have been conducted that have determined the global profile of exosomal miRNAs in plasma of patients with aSAH and healthy controls [145]. An exosomal group of circulating miRNAs that were differently expressed in patients with aSAH (miR-369-3p, miR-136b-3p, miR-410-3p, miR-195-5p, miR-486-3p, and miR-193b-3p) wasidentified. They were expressed significantly differently in patients with aSAH compared to the healthy control group. Four exosomal miRNAs (miR-369-3p, miR-410-3p, miR-193b-3p, and miR-486-3p) showed significantly differentiated expression between the patient group (24 h after aSAH) and the healthy control group. Studies were conducted on mouse models to further confirm the significance of the observed differences in the expression of exosomal miRNAs.Plasma expression of miR-193b-3p remained statistically significant relative to controls in the mouse model of aSAH, while miR-193b-3p expression levels were lower than controls in aSAH mouse brain tissues. It has been suggested that RVG exosomes containing miR-193b-3p may alleviate neuritis by inhibiting HDAC3 expression and activity and increasing NF-κB p65 acetylation after aSAH in vitro.Research must continue to elucidate the benefits of exosomal miR-193b-3p when used as a biomarker.

Exosomes are a natural nanocarrier and intercellular messenger that play a key role in regulating intercellular communication. They can transport both proteins and nucleic acids, including fragments of DNA, mRNA, miRNA, and other non-coding RNAs. Since exosomes were first identified in the 1980s, they have been of interest to scientists around the world, especially in the last few years. A growing body of evidence indicates that miRNAs released by exosomes play a key role in intercellular communication. They affect cell migration and invasiveness, angiogenesis, metastasis, and drug resistance and are also involved in the modulation of the immune response directed against cancer. In addition, due to the presence in the body fluids of patients, they can also be potential diagnostic or prognostic biomarkers.

In the case of personalized medicine, it makes the most sense to develop molecular diagnostic tests based on exosomal miRNAs. To date, the main focus has been on oncology or diagnostics of the ego nervous system using exosome-based technology platforms (http://www.ExosomeDx.com; http://www.HansaBioMed.eu, accessed on 1 January 2014). Exosome populations in biological fluids are heterogeneous and can be derived from all cell types, especially blood cells. Taking this into account, future studies should determine the origin of exosomes present in biological fluids. In addition, it is necessary to consider whether the miRNA levels of exosomes correlate with the specific cellular components of IA. Exosomal miRNAs are currently a benchmark in the search for new therapeutic strategies in the fight against the effects of subarachnoid hemorrhage to improve the treatment of patients with IA.

## 6. The Clinical Significance of miRNA—Future Perspective and Limitations

A biomarker is an objectively measurable biological indicator, indicating or likely to indicate the presence of a disease state or physiological or mental disorders; it is also used to control the body’s response to therapeutics [146].

The ideal biomarker is characterized by high specificity and sensitivity, detectable with minimal use of invasive procedures. The concentration of the biomarker should indicate a medical condition. Circulating miRNAs can be used in diagnostics to assess the state of the disease, providing information on its course. They can also determine the effectiveness of treatment when comparing experimental groups with control groups. The use of the NGS technique has made it possible to simultaneously study thousands of potential biomarkers in the form of circulating miRNAs. This allowsthe detection of biosignatures of the expression of specific diseases, including IA. Circulating miRNAs can help assess the risk of IA rupture. In addition, they can provide information about the molecular mechanism underlying late cerebral ischemia (LCI) resulting from vasoconstriction. Some biomarkers are currently available to clinicians, while others are undergoing preclinical and clinical trials. A notable example is TAmiRNA: circulating platelet miRNAs (thrombomiRs^®^). These are new biomarkers for internal treatment andplatelet reactivity. These markers can be detected in plasma and serum because they are secreted from platelets during their activation. Studies are being conducted that can provide invaluable information about the path of miRNA biomarkers from their discovery to clinical use [147]. The ease of use and clinical uptake of circulating miRNAs areinfluenced by the type of sample; for example, less invasive samples such as blood are often preferred over cerebrospinal fluid.However, the use of these less-invasive types of samples depends on the type of disease. miRNAs found in cerebrospinal fluid may be useful in diagnosing the central nervous system, but may not be useful for tumors in other areas of the body [148]. For the identification of biomarkers, whole blood, serum, and plasma are used. A serious limitation in the use of circulating miRNAs as biomarkers in disease pathology and clinical practice is the lack of acceptable control genes to normalize the data. As endogenous control of circulating miRNAs, numerous endogenous miRNAs (e.g., hsa-miR-16) and small nuclear RNAs such as RNU6B and RNU48 seem to be the most useful [149]. However, their expression may vary depending on the degree of pathology and the stability of RNU6B and RNU48 in biological fluids [150,151]. Quantitative measurement of circulating miRNA can be high due to the origin of miRNAs (for example, from exosomes), which generates conflicting results.Circulating miRNAs in plasma or serum can be contaminated with miRNAs from other blood cells. Therefore, the results require careful interpretation. For research on the specificity of tissue miRNA, it is necessary to use animal models.Comparing patient outcomes with animal test results will minimize false positive results and determine the likely role of the miRNA of interest.Such studies will make it possible to find SAH biomarkersthatshould distinguish between hemorrhages caused by rupture of IA and non-IA hemorrhage. Well-designed and large-scale prospective studies can test the diagnostic usefulness of circulating miRNAs in the early diagnosis of IA or aSAH prediction. Studies should evaluate the effect of confounding factors and other conventional diagnostic and prognostic biomarkers in models to improve IA rupture risk stratification. It is very important to understand the processes that control the release and stability of miRNAs. The correlation between circulating miRNAs and tissue is still unclear. There is a growing body of data that does not support the hypothesis that circulating miRNA levels reflect specific changes occurring in IA tissue, as miRNAs can also come from other cells, such as immune cells. The work presented in the review leads to the conclusion that the expression of circulating miRNAs is significantly altered compared to healthy controls in patients with cracked and unruptured IA. This confirms the alleged usefulness of circulating miRNA as a potential biomarker for identifying IA.Differentiated expression of circulating miRNA levels in bioliquids in IA patients with and without daughter vesiclesis possible. Thus, the cellular and molecular processes underlying initiation, growth, and cracking and IA can occur in different phases. Circulating miRNA profiles are a means by which it is possible to better distinguish between aneurysms that are unlikely to rupture from these unstable IAs. At the moment, fully validated diagnostic tests based on miRNA are available for diseases such as cancer. There is therefore a need for future scientific research to establish the role of circulating miRNAs as diagnostic, predictive, and prognostic biomarkers in IA and aSAH [152].

## 7. Conclusions

Aneurysmal subarachnoid hemorrhage is a catastrophic event that occurs when an intracranial arterial aneurysm ruptures, resulting in the release of oxygenated blood into the subarachnoid space. Untreated subarachnoid hemorrhages are the cause of high mortality, which is why it is so important from a clinical point of view to quickly make a diagnosis and implement appropriate treatments. miRNAs are highly promising diagnostic and prognostic biomarkers in the clinical treatment of patients with IA and aSAH. The number of studies on the clinical value of circulating miRNAs in the diagnosis and prognosis of IA and SAH is growing rapidly. The great interest in miRNAs indicates their possiblyimportantdiagnostic usefulness.

## Figures and Tables

**Figure 1 jcm-11-04630-f001:**
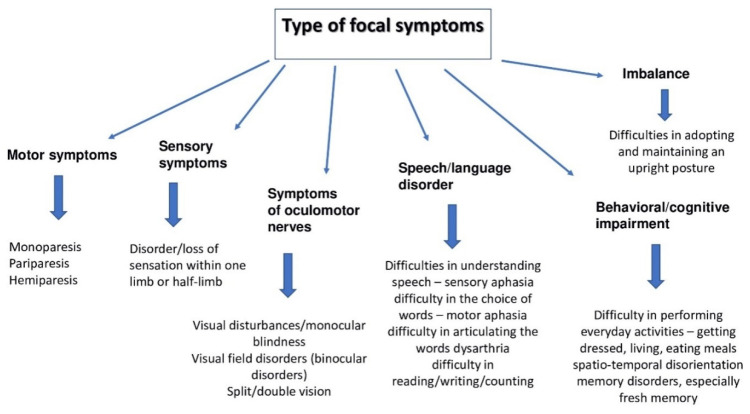
Focal symptoms occurring during subarachnoid hemorrhage.

**Figure 2 jcm-11-04630-f002:**
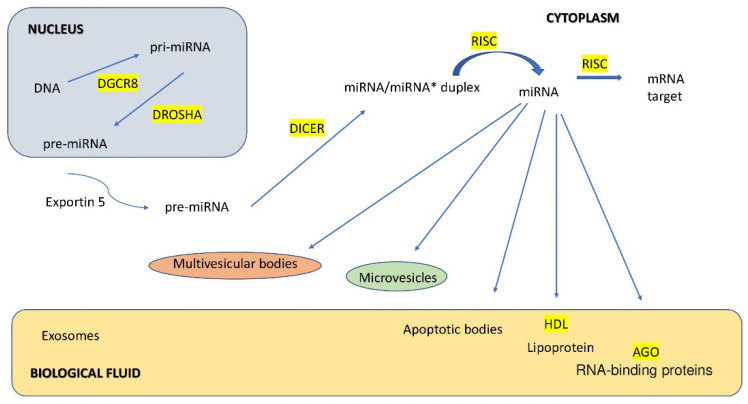
miRNA biogenesis.

**Table 1 jcm-11-04630-t001:** Classification of the severity of subarachnoid hemorrhage.

Grade	Symptoms
I	No symptoms, possible presence of neck stiffness
II	Severe headaches, neck stiffness, cranial nerve paralysis
III	Confusion, drowsiness, slight focal symptoms
IV	Stupor, hemiparesis, decerebrate rigidity, vegetative disorders
V	Deep coma, decerebrate rigidity, moribund

**Table 2 jcm-11-04630-t002:** WFNS scale.

Grade	Glasgow Scale	Movement Disorders
I	15	Absent
II	14–13	Absent
III	14–13	Present
IV	12–7	Present or not
V	6–3	Present or not

**Table 3 jcm-11-04630-t003:** Fisher scale.

Grade	Amount of Blood in Computed Tomography
I	Lack of blood in the computed tomography image
II	Leaked or thin layer of blood < 1 mm thick
III	Localized clot and/or blood layer > 1 mm thick
IV	Intraventricular and/or interstitial blood clot

**Table 4 jcm-11-04630-t004:** Circulating miRNAs as biomarkers in IA and aSAH.

	miRNA	Regulation
Diagnostic of IAs or aSAH	miR-1297	Up [106]
	miR-502-5p	Down [107]
	miR-4320	Up [108]
	miR-143	Down [101]
	miR-145	Down [101]
	miR-155	Up [102]
	miR-29a	Down [109]
	miR-200a-3p	Up [110]
	miR-let7-b	Down [110]
	miR-16	Up [96]
	miR-25	Up [96]
	miR-15a-5p	Up [104]
	miR-146-5p	Down [104]
	miR-126	Up [111]
	miR-132-3p	Up [112]
Prognostic of aSAH	miR-1297	Up [106]
	miR-502-5p	Down [107]
	miR-29a	Down [109]
	miR-200a-3p	Up [110]
	miR-146-5p	Down [104]
	miR-92a	Down [113]
	let-7b	Down [113]
	miR-3177-3p	Up [114]
	miR-132-3p	Up [112]
	miR-15a	Up [115]

## Data Availability

Data sharing is not applicable to this article.

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
