# Peer review of "microRNAs in Subarachnoid Hemorrhage (Review of Literature)"

_jcm, 2022, doi:10.3390/jcm11154630_

Round 1

Reviewer 1 Report

This is a timely and important review on the role of miRNAs in subarachnoid hemorrhage, albeit I would recommend widen the scope for a broader readership (see specific points below).

The article reads well and the literature appears carefully selected.

While I find the clinical part meant to give essential background info a bit lengthy (here and there even covering aspects that one would not expect to find in an article that primarily centers around miRNAs e.g. interview etc), I felt that the description of pathophysiological mechanisms and their association to miRNAs falls short. Thus the described examples of miRNAs and their relation to SAH are merely associations (and not too novel). At best they may serve as diagnostic markers, but not necessarily as therapeutic targets. Therefore, it is important to get this better linked to the primary and/or secondary pathophysiology in SAH.

This is also very relevant as it is currently very unlikely that miRNAs make it into routine laboratory diagnostics, and hence even though being diagnostically or prognostically informative, the mere focus on diagnostic aspects may be too narrow.

As one example (several other links exist as well): it is relatively well established that components of the hemostatic system play a crucial role in the pathogenesis and modulation of brain disorders including haemorrhage (for example PMID: 26761005). Why is this relevant in this context? miRNAs in turn are potent regulators of the hemostatic system (PMID: 30207063) and are therefore considered important modulators of this system (locally and systems-wide) in response to all kinds of events; most notably bleeding (PMID: 32898547). In addition, miRNAs travel with the bloodstream, and can ultimately show up in SAH (or the cerebrospinal fluid) as well. At the same time, miRNAs show up in these fluids (and blood stream) as a result of neuronal tissue destruction.

Therefore, cause and consequence of miRNA expression pattern must be clearly distinguished – especially when it comes to therapeutic targeting of miRNAs.

I expect integrating and elaborating this a bit further beyond mere associations gives novelty and warrants high interest in this paper.

Specific points:

The introductory chapter on miRNAs (lines 234-353) is very long (with limited novelty), while the following chapter requires to be expanded a bit further (see main comment above).

Line 230: Non-coding RNAs comprise a much larger family than just miRNAs. This needs to be rephrased.

Line 315: I guess it should read “Drosha”

Line 362: What are “microRNA disorders”. This wording can be misunderstood

Line 487: there is a typo “….possible. P…..”

I missed some papers that give an outlook on the physiological role of miRNAs in the brain (a.g. PMID: 25053999) and ultimately the therapeutic punchline (e.g. PMID: 19763905)

Author Response

Thank you for your review.

I would like to kindly ask you to reconsider the publication of our revised paper:

" microRNAs (miRNAs) in subarachnoid hemorrhage (SAH) (review of literature).”.

I hereby provide responses to the reviewers and list the changes that have been made in the revised version of our paper.

Thank you for your comments. The article has been corrected.  We've added a chapter on clinical significance miRNA. We have also added a chapter on miRNA in naurological diseases, taking into account the work of PMID: 25053999) and PMID: 19763905). It seems to us that we have aligned the proportions between the chapters (description of miRNA and meaning in SAH, IA)

I hope you find our revised Manuscript satisfying so that it can meet the criteria of publication in your Journal.

Looking forward to hearing from you,

Yours sincerely,

Beata Smolarz

Reviewer 2 Report

The manuscript is interesting to read, however I see the following issues that should be resolved before publishing this paper.

  1. Please use full name when they appear in the manuscript for the first time, for example IA.
  2. Please keep every consistent in the manuscript, like miRNA and MiRNA.
  3. ‘that can be used in the clinic to help diagnose and segregate patients. Further clinical 514 trials of these circulating miRNA panels could improve care for stroke patients.’ is this a separate paragraph?
  4. The clinical significance of in this study should be described in detail.
  5. Could you add some Figures to illustrate your findings?

Author Response

Thank you for your review.

I would like to kindly ask you to reconsider the publication of our revised paper:

" microRNAs (miRNAs) in subarachnoid hemorrhage (SAH) (review of literature).”.

I hereby provide responses to the reviewers and list the changes that have been made in the revised version of our paper.

The manuscript is interesting to read, however I see the following issues that should be resolved before publishing this paper.

Thank you for your comments. The article has been corrected.  We've added a chapter on clinical significance and put the figure in chapter 2.

I hope you find our revised Manuscript satisfying so that it can meet the criteria of publication in your Journal.

Looking forward to hearing from you,

Yours sincerely,

Beata Smolarz